# Beneficial Effects of Zoledronic Acid on Tendons of the Osteogenesis Imperfecta Mouse (Oim)

**DOI:** 10.3390/ph16060832

**Published:** 2023-06-02

**Authors:** Antoine Chretien, Guillaume Mabilleau, Jean Lebacq, Pierre-Louis Docquier, Catherine Behets

**Affiliations:** 1Pole of Morphology, Institute of Experimental and Clinical Research, Université Catholique de Louvain, 1200 Brussels, Belgium; 2Univ Angers, Nantes Université, Oniris, Inserm, UMR_S 1229—RMeS, REGOS, SFR ICAT, F-49000 Angers, France; 3Centre Hospitalier Universitaire d’Angers, Department of Cell and Tissue Pathology, Bone Pathology Unit, F-49000 Angers, France; 4Institute of NeuroScience (IoNS), Université Catholique de Louvain, 1200 Brussels, Belgium; 5Neuromusculoskeletal Lab, Institute of Experimental and Clinical Research, Université Catholique de Louvain, 1200 Brussels, Belgium

**Keywords:** osteogenesis imperfecta, oim, tendon, bone–tendon unit, zoledronic acid

## Abstract

Osteogenesis imperfecta (OI) is a genetic disorder of connective tissue characterized by spontaneous fractures, bone deformities, impaired growth and posture, as well as extra-skeletal manifestations. Recent studies have underlined an impairment of the osteotendinous complex in mice models of OI. The first objective of the present work was to further investigate the properties of tendons in the osteogenesis imperfecta mouse (oim), a model characterized by a mutation in the *COL1A2* gene. The second objective was to identify the possible beneficial effects of zoledronic acid on tendons. Oim received a single intravenous injection of zoledronic acid (ZA group) at 5 weeks and were euthanized at 14 weeks. Their tendons were compared with those of untreated oim (oim group) and control mice (WT group) by histology, mechanical tests, western blotting and Raman spectroscopy. The ulnar epiphysis had a significantly lower relative bone surface (BV/TV) in oim than WT mice. The tendon of the triceps brachii was also significantly less birefringent and displayed numerous chondrocytes aligned along the fibers. ZA mice showed an increase in BV/TV of the ulnar epiphysis and in tendon birefringence. The tendon of the flexor digitorum longus was significantly less viscous in oim than WT mice; in ZA-treated mice, there was an improvement of viscoelastic properties, especially in the toe region of stress-strain curve, which corresponds to collagen crimp. The tendons of both oim and ZA groups did not show any significant change in the expression of decorin or tenomodulin. Finally, Raman spectroscopy highlighted differences in material properties between ZA and WT tendons. There was also a significant increase in the rate of hydroxyproline in the tendons of ZA mice compared with oim ones. This study highlighted changes in matrix organization and an alteration of mechanical properties in oim tendons; zoledronic acid treatment had beneficial effects on these parameters. In the future, it will be interesting to better understand the underlying mechanisms which are possibly linked to a greater solicitation of the musculoskeletal system.

## 1. Introduction

Osteogenesis imperfecta (OI) is a genetic disorder of connective tissue generally associated with pathogenic variants in *COL1A1* or *COL1A2* genes. They encode the pro-alpha-1 and alpha-2 chains of the type I collagen triple helix, whose main function is to provide tensile strength to tissues [1]. Defects in the collagen structure lead to low-trauma fractures, bone deformities and impaired growth [2].

The diagnosis of OI is currently based on clinical and radiological arguments. The great variability of manifestations leads to variable circumstances of discovery and evolution [3]. Recent advances in genetics have shown that the deficiency of a protein involved in collagen synthesis or post-translational modification, as well as in regulation of bone-forming cells, also generates rare forms of OI [4].

Patients require multidisciplinary care, including medical treatment, surgery and physiotherapy. The main pharmacological treatments used are anti-resorptive agents [5]. These drugs include bisphosphonates which aim to inhibit osteoclast activity through the mevalonate pathway and thereby increase bone mass [6]. Among bisphosphonates, zoledronic acid (ZA) [1-hydroxy-2-(1H-imidazol-1-yl)-1-phosphonoethyl] belongs to nitrogenous ones, and it is characterized by the presence of a nitrogen atom in the R2 chain. In OI, it decreases the number of bone fractures, reduces pain and improves functional status in children [7]. It is one of the most widely used for intravenous injections in OI, along with pamidronate [8].

In addition to bone defects, OI patients also present extra-skeletal manifestations such as dentinogenesis imperfecta, blue sclera, hearing impairment, joint hypermobility and more rarely, muscle weakness [9]. Several case reports have notably described tendon ruptures in patients [10,11,12,13]. Recent publications have underlined an impairment of the tendon–to–bone unit in different mice models of osteogenesis imperfecta [14,15]. The oim is a validated model of human type III OI, a severe form of the disease with progressive deformity. This model is characterized by a mutation in the *COL1A2* gene, a guanine deletion in the C-terminal position which induces a shift of the last amino acids [16]. Consequently, the α-2 chain is not able to participate in the triple helix formation. The resulting production of α1(I) homotrimers in the homozygous (oim/oim) mouse leads to short stature, low bone mass, skeletal deformities and fragile bones [17]. In young mice, tendons present alterations of their biomechanical, microstructural and material properties. The tendons are less rigid and less resistant than WT ones. They also present a significantly higher rate of pentosidine, without significant modification in enzymatic cross-linking [18].

Tendons are made up of dense regular connective tissue which transfers force from muscles to bones. The organic extracellular matrix is synthesized by tenocytes and mainly composed of type I collagen. Other types of collagens and non-collagenous proteins such as decorin, the most abundant proteoglycan in tendon, and tenomodulin, a glycoprotein known as a marker of tendon maturation [19], are also present in low amounts. The viscoelastic properties of the tendon strongly depend on the structure and organization of the collagen. Several publications have highlighted that the collagen network, cross-linking (whether enzymatic or non-enzymatic) [20] and collagen bound-water [21] play a key role in the mechanical properties of bone.

The insertion of tendon into bone, or enthesis, is a territory of junction between two different tissues and functions as a stress-reducer [22], i.e., a unit whose properties adapt according to the constraints it receives [23]. To date, little is known about the collagen properties in OI tendons and no published study has investigated the possible effects of bisphosphonate administration on this tissue. The objectives of this work were to identify OI-specific changes in the tendons of 14-week-old female oim and to assess potential beneficial effects of a treatment with zoledronic acid. The data were compared with age- and sex-matched oim and WT littermates.

## 2. Results

The results are subdivided into different parts according to the objectives of this work. First, a macro- to microscopic analysis and an investigation of mechanical properties will be described. Secondly, tissue composition and material properties will be analyzed with Raman spectroscopy and western blotting.

### 2.1. Macro- to Microscopic Analysis of Bone and Tendon

At 14 weeks of age, BV/TV of the ulnar epiphysis was significantly lower (−41%) in oim than WT mice (Table 1). This parameter was increased in ZA-treated mice (+25%), which confirms the positive effect of ZA on bone quantity (Figure 1A). Nevertheless, the values were still significantly lower than WT ones (−21%). At 5 weeks, oim and ZA mice presented calcaneus fractures (Figure 1B) with, on average, more than one calcaneal fracture per mouse (1.44 ± 0.13). Half of them had a fracture or a deformation on both sides. There was a slight increase at the end of the experiment since two additional fractures were found in oim and one in the ZA group. No fractures were observed in WT mice during the whole experiment.

Lower limb imaging at 14 weeks showed calcifications of the triceps surae tendon (Figure 1B) in 40% of oim and 53% of ZA mice. No such calcifications were visible in other locations, neither at 5 weeks, nor in WT mice. Aligned chondrocytes were observed in these islets of calcification (Figure 1C), as well as in the tendon itself. These alignments of successive chondrocyte-like cells in the tendon were also systematically found in the triceps brachii of oim (Figure 1D). Several cells were observed in the other groups but were much less numerous.

The cross-sectional area of the FDS tendon was significantly lower in oim than WT mice (*p* < 0.0001); ZA treatment did not modify this area. The respective linear regressions between tendon cross-section and ulna length tended to be similar in oim and ZA groups but were lower than in WT ones (Figure 1E).

Finally, the microscopic analysis of histological sections under polarized light showed that oim tendons were less birefringent than WT ones (Figure 1F). Indeed, their mean grey level was significantly lower (−55%). The tendons of treated mice presented equivalent grey levels to WT ones and significantly higher values than oim ones (Table 1). Consequently, oim and WT mice present a clear difference in tendon collagen organization while tendons of ZA mice seem similar to controls.

### 2.2. Mechanical Tests

Tensile tests until rupture showed that oim FDL tendons had a lower ultimate stress (−30%) and toughness (−31%) than WT ones, but the differences were not significant. Tendons of ZA mice presented intermediate values for these markers of mechanical resistance. The elastic modulus of ZA tendons was larger than oim ones (+68%) and comparable to WT ones (Table 2). In oim tendons, the stress at the end of the toe region was significantly lower than in WT ones (−49%) but the strain was around 4% of deformation for both groups (Figure 2A). In this part of the stress-strain curve, tendons of ZA mice presented similar value as WT ones, which show beneficial effects of the treatment.

The stress relaxation of oim FDL tendons was significantly lower than that of WT ones (−51%), which means that oim tendons were less viscous (Figure 2B). Treated mice presented a significant improvement of this parameter (Table 2).

### 2.3. Tissue Composition and Material Properties

The expression of decorin looked similar in the tail tendon of all groups (Figure 3). Consequently, neither the pathology nor the treatment seemed to have an impact on proteoglycan contents in the tendons. Concerning tenomodulin, its expression was slightly lower in oim than in other groups but without significance.

Furthermore, Raman spectroscopy analysis did not show clear differences in tissue composition and material properties between WT and oim triceps brachii tendons. The rate of enzymatic cross-linking was lower in oim (−12%) but without significance (Table 3). Nevertheless, ZA tendons presented significantly higher values for pentosidine and porosity than WT ones (respectively +33% and +63%). The rate of hydroxyproline was also significantly enhanced in ZA treated mice compared with oim ones (+35%). Finally, glycosaminoglycans rates were very similar in all groups.

## 3. Discussion

This study highlights original features of the bone–tendon units of oim as well as some beneficial effects of zoledronic acid on this osteotendinous complex. Most of the observations made in 14-week-old oim were similar to the results obtained in younger ones [18].

The slight difference in calcaneus fracture rates between groups at 5 weeks was only due to their random distribution since mice of the ZA-treated group had not yet received the injection. This bone looks particularly frail in oim during the first weeks of life, as evidenced by the very high fracture rate. Although ZA administration improved BV/TV of the ulnar epiphysis, it was not possible to highlight an effect of treatment on fracture count due to the small augmentation of fracture or deformation at the end of the experiment.

The islets of calcified fibrocartilage into the triceps surae tendons were found in 53% of ZA mice versus 40% of oim ones but the difference was not significant. These calcifications were also described in another study involving the oim [23]. Their origin is not clearly defined but previous publications showed that over-use tendinopathy induced differentiation of tendon stem cells into chondrocytes, osteocytes and adipocytes [24,25]. Moreover, in the case of tendon injury, the recruited progenitor cells have a strong chondrogenic potential [26]. Taken together, these elements could be indicators of tendon damage.

As observed before, non-collagenous protein did not seem to be impacted although the expression of tenomodulin was slightly lower in oim than in the other two groups. Consequently, the main differences observed can be explained by modifications of the collagen matrix.

The tensile tests give very interesting information on oim tendon biomechanics, which have never been precisely described. The viscosity plays an important role in mechanic properties since it increases the resistance to lengthening [27] and allows greater transfer of force to the bone unit. As a conclusion, the combination of lower viscosity and stiffness suggests that the force transfer from muscles to bones is less efficient in oim tendons. The tendon ruptures described in OI patients do not seem to be explained by a lack of extensibility since no difference was found for the ultimate strain.

The Raman spectroscopy did not highlight a significant difference in the rate of enzymatic cross-links between oim and WT tendons, as previously observed in young oim [18]. However, the rate of hydroxyproline was significantly enhanced in ZA tendons compared with oim ones. These residues are known to play an essential role in collagen triple helix stability [28]. Finally, significant changes were found between ZA and WT groups. Porosity is the space filled with free water during a lifetime and was higher in ZA tendons than in control ones. Pentosidine is the most frequently characterized non-enzymatic cross-link and tends to increase with age and disease [29]. This non-enzymatic cross-link results from a glycation process forming advanced glycation end products (AGEs). They are mainly accumulated in collagen because of its low turnover and they affect the tissue mechanical properties [30]. Increased pentosidine rate in ZA tendons is undesirable but the emergence of antiglycating compounds could be a fruitful approach to deal with it [31,32].

A recent gait analysis in OI revealed that children who had received more bisphosphonates injections had less kinematic impairment [33]. In our work, treatment with ZA significantly improved BV/TV in the ulnar epiphysis, which confirmed the efficient administration of the treatment. Due to this positive effect, we can consider that, with stronger bones, mice are more mobile and gain greater use of their musculoskeletal system. This could partly explain the positive effects of treatment on tendon organization. Indeed, several publications have showed beneficial effects of mechanical stimulation on cell culture since it induces the expression of tendon-specific genes [34] and promotes tenogenic differentiation [35]. Stem progenitor cells play key roles in tendon development, homeostasis and pathogenesis through the TGF beta superfamily [36] and are strongly regulated by mechanical loading [37].

The alteration of toe region on stress-strain curves suggested a change in the collagen crimp of oim tendons. This hypothesis is supported by the histological images of birefringence since quantitative polarized light microscopy is also an efficient technique to analyze the crimp characteristics of collagen [38]. The beneficial effects of ZA were observed in both methods and another study previously showed changes in collagen birefringence of tendon after certain types of physical training [39].

Although the tendon is often considered a simple force transmitter, our work shows that it is an active tissue, capable of adapting. In human, physical training could efficiently improve the Achilles tendon mechanical properties, in particular eccentric exercise [40]. This reinforces the importance of physiotherapy for OI patients, in order to act on all components of the musculoskeletal system.

Some mechanisms and properties of tendons are still being discovered; it has been shown recently that intrafibrillar collagen mineralization can generate contractile stress in tendon [41] or that changes in the osmotic pressure surrounding the collagen molecules should produce macroscopic stresses [42]. A better understanding of this hard-to-treat tissue is still necessary to improve patient care.

Although bisphosphonate therapy remains the main pharmaceutical treatment for OI [8], side effects have been widely reported in patients such as systemic inflammatory reaction, renal failure, ocular complications, nephrotic syndrome and osteonecrosis of the jaw [43]. In the experimental field, only one animal study with Wistar rats has focused on the effects of zoledronic acid on the bone–tendon unit. While our experimental designs are strongly different, a reduction in osteotendinous stiffness was also observed [44].

In OI mouse models, an accumulation of mutated type I collagen in the endoplasmic reticulum of osteoblasts induces stress and may be related to the clinical outcome [45]. A recent study also highlighted this phenomenon in skin fibroblasts [46]. It would be interesting to investigate this phenomenon on tenocytes, and to study a possible effect of mechanical stimulation on this cellular stress.

In conclusion, this study highlights original features of the tendon to bone unit in the oim and demonstrates beneficial effects of zoledronic acid on the osteotendinous complex. The ulnar epiphysis had a significantly lower BV/TV in oim than in the WT group. The tendon of the triceps brachii was also significantly less birefringent and displayed numerous chondrocytes aligned along the fibers. ZA mice showed an increase in BV/TV and in tendon birefringence. The flexor digitorum longus tendon was significantly less viscous in the oim group than in the WT one. The tendons of ZA-treated mice presented an improvement of viscoelastic properties, especially in the toe region of the stress-strain curve, which corresponds to collagen crimp. The tendons of both oim and ZA groups did not show any significant change in the expression of decorin or tenomodulin. Finally, Raman spectroscopy highlighted differences in material properties between ZA and WT tendons. A significant increase was also observed in the rate of hydroxyproline for ZA mice compared with oim ones. In the future, it will be interesting to better understand the underlying mechanisms, which are possibly linked to a greater solicitation of the musculoskeletal system. It also seems necessary to investigate non-skeletal connective tissues such as tendons and ligaments in humans in order to improve OI patient care.

## 4. Materials and Methods

### 4.1. Animals

Homozygous oim (strain *B6C3Fe a/a-Col1a2oim/J*) and wildtype female mice (strain B6C3Fe-a/a +/+; SN 1815) were used for this study (Charles River Laboratories, L’Arbresles, France). The genotype was monitored by polymerase chain reaction, using the primers *ggctttcctagaccccgatgcttag* as forward; *gtcttgccccattcatttgtt* as OI reverse; and *gtcttgccccattcatttgtc* as WT reverse. All experiments are in accordance with the Belgian federal law. The ethics committee for animal research of the Université Catholique de Louvain also approved the study (reference 2020/UCL/MD/047). The mice were housed at 24 °C with a 12/12 h light-dark cycle.

### 4.2. Procedures

At five weeks, twenty female oim were randomly assigned to oim or zoledronic acid-treated (ZA) groups. Zoledronic acid was administered once by intravenous injection (100 µg/kg). A last group with ten age- and sex-matched WT mice was also used. All mice were weighed and scanned at 5, 9 and 14 weeks with an in vivo CT (Nano SPECT/CT, Mediso) under sevoflurane inhalation anesthesia. Thereafter, they were euthanized by sevoflurane overdose inhalation and dissected in order to take elbow, forearm, tail tendons and both flexor digitorum longus tendons. The ulna length and calcaneus fracture count were calculated with the scans. It should be mentioned that the effects of ZA on long bone fractures will be of concern in another experiment. This study only focuses on tendon insertion sites and their avulsion fractures.

### 4.3. Histology

The elbow and forearm were fixed in 4% formaldehyde and decalcified before embedding in paraffin and sectioning. The 5-micron thick slices were stained with picro-sirius red and observed under optical microscope to measure the relative bone surface (BV/TV) in the ulnar epiphysis. Unstained sections of triceps brachii tendon were observed under polarized light microscopy to study collagen organization. Anisotropic materials have a preferential orientation allowing an optimal reaction of the refracted beam with the sample. Appearing as a brightness in the image, the birefringence was quantified by comparing grey values from 0 (black) to 255 (white). Finally, unstained transversal sections through the distal radio-ulnar joint allowed measurement of the cross-section area of the flexor digitorum superficialis (FDS) tendon. All histological sections were imaged with the Axioplan microscope (ZEISS) and digitalized using the Nikon Digital sight DS SMC camera with NIS-Element BR 3.0 software. The different measurements were performed with ImageJ software (version 1.52A).

### 4.4. Mechanical Test

The mechanical properties of the flexor digitorum longus tendons were evaluated with tensile tests. One end of the tendon was connected to a length transducer and an electromagnetic motor, and the other one to an isometric force transducer. The tendon was kept immersed in phosphate-buffered saline throughout the measurements to avoid any drying which could affect the results. Length and force signals were recorded using Picoscope6 software (Pico Technology, St Neots, UK, www.picotech.com). The left tendon underwent sinusoidal stimulation at a frequency of 10 Hz and with a deformation of 10% from the initial length. The right one was evaluated until rupture as previously described [18]. The first part of the stress-strain curve, before linear relation, is due to the collagen crimp and is called the toe region. The elastic modulus is measured by the slope of the linear part of the curve and the toughness of the tendon corresponds to the area under the curve. The ultimate point is located where the rupture occurs, and the ultimate stress and strain are determined at this point. To define the limit of the toe region on stress-strain curve, a line is generated with the slope and intersecting the inflection point [47]. The second derivative of the curve is used to calculate the inflection point. With the prolonged maintenance of the deformation in time, the difference of stress between the beginning and the end of the plateau can be calculated and is proportional to the viscosity.

### 4.5. Western Blotting

The tail tendons were homogenized with a radioimmunoprecipitation buffer containing a protease inhibitor and PhoSTOP (Roche, Basel, Switzerland). The concentration of proteins was calculated with a bicinchoninic protein assay kit (ThermoFischer Scientific, Waltham, MA, USA). Eight percent sodium dodecyl-sulfate polyacrylamide gels were prepared, and the wells were loaded with 20 µg of proteins and the molecular weight marker (PageRuler Prestained, ThermoFisher Scientific, 26614). The migration was performed in migration buffer (tris-ethylenediaminetetraacetic acid, sodium dodecyl-sulfate, glycine) under 120 V for 90 min. Proteins were transferred onto nitrocellulose blotting membranes (GE Healthcare, Chicago, IL, USA) with an amperage of 350 mA for 90 min. Membranes were blocked for one hour with tris-buffered saline containing 0.1% Tween and 5% Bovine Serum Albumin. The primary antibodies, decorin (Abcam, ref: ab137508, dilution 1/500) and tenomodulin (Abcam, ref: ab203676, dilution 1/500), were then incubated at 4 °C overnight. After washing, membranes were incubated with the secondary antibody for 1 h and then developed on an X-ray film (CL-X Posure^®^ films, ThermoFischer Scientific, Waltham, MA, USA) using Electrochemiluminescence Western Blotting Detection Reagents (GE Healthcare, Chicago, IL, USA). Protein levels were normalized to β-actin (Sigma, ref: A5441, dilution 1/50,000). Films were scanned and quantified by densitometry using ImageJ software (Version 1.52A). In results, the image is cut off since the third column was an ultimately unexploited group, but the measurements were correctly made on the same gels.

### 4.6. Raman Spectroscopy

Raman spectroscopy is a non-destructive vibrational technique using a monochromatic laser beam. Scattered light produced by molecular vibrations reveals bands at specific frequencies in the Raman spectrum and allows analysis of the composition of a sample. The paraffin blocks of elbows were trimmed, and the triceps brachii tendon composition was studied using a Renishaw InVia Qontor microscope. This was completed by using a 785 nm monochromatic laser diode (Renishaw Plc, Wotton-under-Edge, UK) with a laser power of 10 mW and a 20× objective. Three spectra were recorded for each tendon with an integration time of 5 s and 15 accumulations. Enzymatic collagen cross-linking (intensity ratio 1670/1690 cm^−1^), non-enzymatic collagen cross-linking represented by the pentosidine content (Intensity ratio 1345/920 cm^−1^), hydroxyproline content (Intensity ratio 872/920 cm^−1^), glycosaminoglycan content (intensity ratio 1380/920 cm^−1^) and nanoporosity (Intensity ratio 1296/920 cm^−1^) were computed. The spectra were processed with MATLAB software (The MathWorks, Inc., Natick, MA, USA).

### 4.7. Statistics

Statistical analysis was performed using ANOVA, or the non-parametric equivalents (Prism 5.0 for Windows, GraphPad Software, Boston, MA, USA, www.graphpad.com). Pairwise comparisons were performed using Bonferroni’s test. Data are presented as means ± standard error of the mean (SEM). Differences were considered significant at *p* < 0.05.

## 5. Conclusions

This study highlighted changes in matrix organization and an alteration of mechanical properties in oim tendons; zoledronic acid treatment had beneficial effects on these parameters. In the future, it will be interesting to better understand the underlying mechanisms which are possibly linked to a greater solicitation of the musculoskeletal system.

## Figures and Tables

**Figure 1 pharmaceuticals-16-00832-f001:**
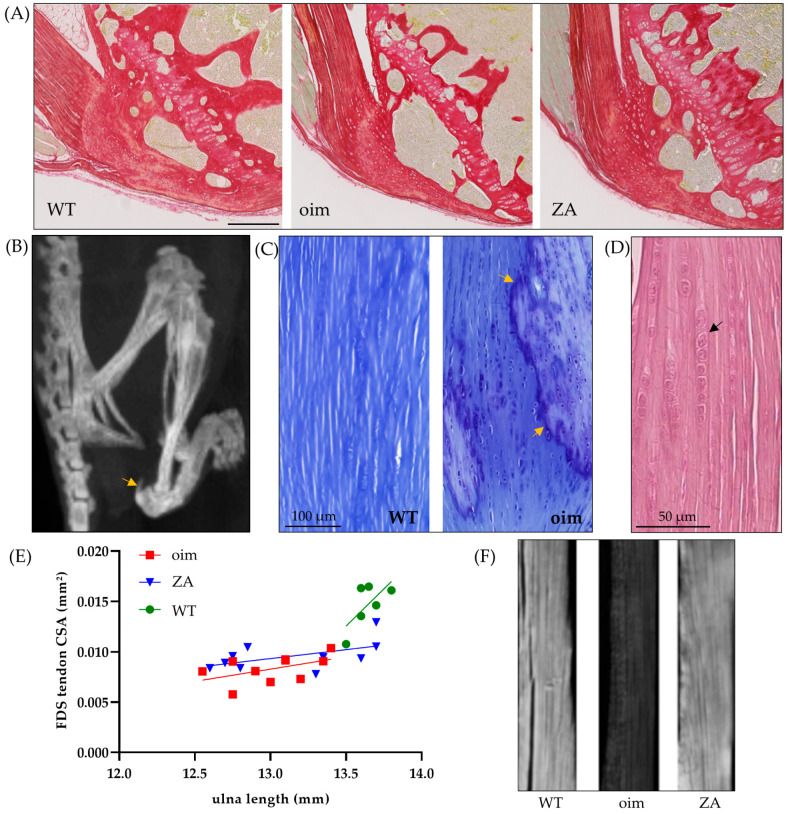
(**A**) Sagittal section through the ulnar epiphysis, sirius red staining; the scale bar is 200 µm. (**B**) Lower limb imaging with in vivo CT in a 14-week-old oim; the arrow shows calcaneus avulsion and tendon calcification. (**C**) Sagittal section through the triceps surae tendon, toluidin blue staining; the arrows show the calcification front. (**D**) Sagittal section through triceps brachii tendon in an oim, hematoxylin eosin staining; the arrow shows the successively aligned chondrocytes. (**E**) Comparison of linear regressions between cross-section area of flexor digitorum superficialis tendons (FDS) and ulna length. R^2^ = 0.25 for oim, R^2^ = 0.28 for ZA, R^2^ = 0.46 for WT; slope: *p* = 0.14, y-int: *p* < 0.001. (**F**) Sagittal section through triceps brachii tendon, polarized light.

**Figure 2 pharmaceuticals-16-00832-f002:**
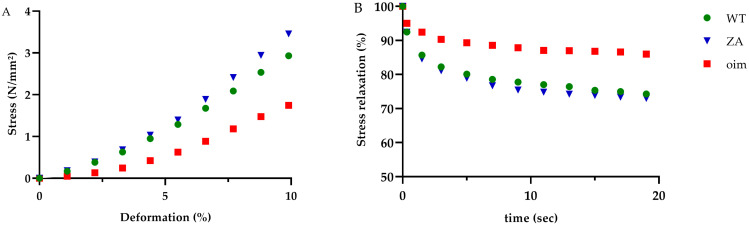
(**A**) Stress strain curve of tensile test of left flexor digitorum longus (FDL) tendons; the deformation was 10% of initial length. (**B**) Evolution of stress relaxation during the maintenance of deformation of right FDL tendons for 20 s, *n* = 8 for oim and *n* = 7 for WT and ZA mice; data are means for each group.

**Figure 3 pharmaceuticals-16-00832-f003:**
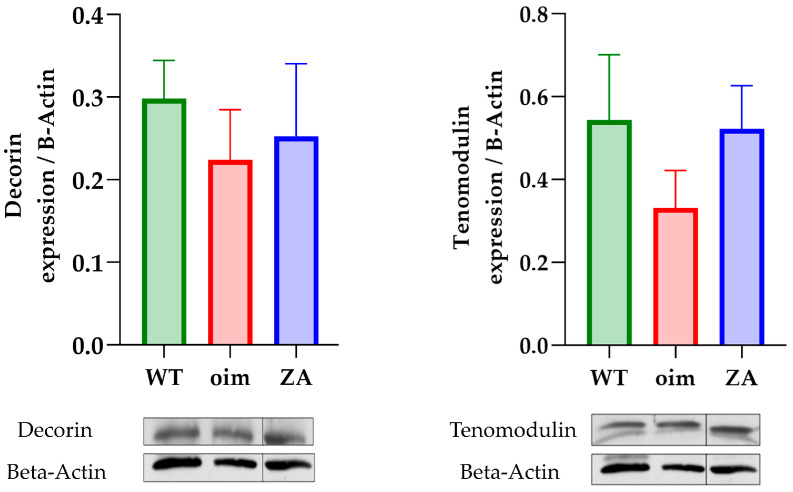
Expression of decorin and tenomodulin in the tail tendon; *n* = 8 for all groups. Results are expressed as mean ± SEM and normalized to beta-actin. The measurements were made on the same gels but the image is cut off since the third column was an ultimately unexploited group.

**Table 1 pharmaceuticals-16-00832-t001:** Relative bone surface of ulnar epiphysis and birefringence of triceps brachii tendons of WT, oim and ZA mice.

	WT	Oim	ZA	*p*-Value
Relative bone surface (BV/TV)	0.65 ± 0.05	0.38 ± 0.01	0.51 ± 0.03	*p* < 0.001; a **; b,c *
Mean grey level (birefringence)	89.5 ± 10.2	40.6 ± 4.1	95.0 ± 10.8	*p* < 0.01; a,b **

BV/TV = Bone Volume/Total Volume; grey level from 0 (black) to 255 (white), *n* = 7 for oim, *n* = 8 for WT and *n* = 9 for ZA mice; results are expressed as mean ± SEM; a = oim vs. WT, b = oim vs. ZA, c = WT vs. ZA; * *p* < 0.05, ** *p* < 0.01.

**Table 2 pharmaceuticals-16-00832-t002:** Mechanical properties of the flexor digitorum longus tendon of WT, oim and ZA mice under tensile tests.

	WT	Oim	ZA	*p*-Value
Ultimate stress (N/mm^2^)	14.0 ± 1.7	9.8 ± 1.1	13.0 ± 1.9	0.16
Ultimate strain (%)	46.1 ± 3.6	49.9 ± 2.8	42.7 ± 3.1	0.30
Toughness (N/mm^2^)	313 ± 67	215 ± 29	274 ± 47	0.51
Elastic Modulus (N/mm^2^)	0.31 ± 0.06	0.19 ± 0.02	0.32 ± 0.04	0.09
Stress toe region (N/mm^2^)	0.63 ± 0.11	0.32 ± 0.06	0.58 ±0.07	*p* < 0.05 a,b *
Stress relaxation (%)	27.4 ± 2.3	13.5 ± 1.7	27.9 ± 2.3	*p* < 0.001 a,b **

*n* = 8 for oim and *n* = 7 for WT and ZA mice; results are expressed as mean ± SEM; a = oim vs. WT, b = oim vs. ZA, * *p* < 0.05, ** *p* < 0.01.

**Table 3 pharmaceuticals-16-00832-t003:** Tissue composition and material properties of the triceps brachii tendon of WT, oim and ZA mice assessed with Raman spectroscopy.

Raman Spectroscopy 700–1800 cm^−1^	WT	Oim	ZA	*p*-Value
Enzymatic Collagen Cross-link IR 1670/1690	1.41 ± 0.32	1.24 ± 0.10	1.31 ± 0.25	0.15
Pentosidine (AGE) IR 1345/920	1.91 ± 0.45	2.03 ± 0.20	2.55 ± 0.56	*p* < 0.05 c *
Hydroxyproline IR 872/920	0.73 ± 0.16	0.57 ± 0.11	0.88 ± 0.21	*p* < 0.05 b *
Glycosaminoglycan IR 1380/920	0.07 ± 0.02	0.06 ± 0.02	0.06 ± 0.01	0.90
Porosity IR 1380/920	7.1 ± 1.9	9.6 ± 3.4	11.6 ± 3.0	*p* < 0.05 c *

AGE = Advanced Glycation End products. Results are expressed as mean ± SEM. *n* = 6 for WT, *n* = 7 for oim and *n* = 5 for ZA; b = oim vs. ZA, c = WT vs. ZA, * *p* < 0.05.

## Data Availability

Not applicable.

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
