# Peer review of "Beneficial Effects of Zoledronic Acid on Tendons of the Osteogenesis Imperfecta Mouse (Oim)"

_pharmaceuticals, 2023, doi:10.3390/ph16060832_

Round 1

Reviewer 1 Report

Thank the author for their hard work. But there are many problems in the manuscript. First of all, the picture resolution is not high. The second results show no statistical results. A large number of additional experiments are needed to meet the submission requirements.

Thank the author for their hard work. But there are many problems in the manuscript. First of all, the picture resolution is not high. The second results show no statistical results. A large number of additional experiments are needed to meet the submission requirements.

Reviewer 2 Report

please see the attached file for your information.

The manuscript shoud be further improved.

Reviewer 3 Report

1.      OI is caused by defects in or related to a protein called type 1 collagen. Collagen is an essential building block of the body. The body uses type 1 collagen to make bones strong and to build tendons, ligaments, teeth, and the whites of the eyes. Certain gene changes, or mutations, cause collagen defects.

I think that author could improve the introduction by providing the function, type, and significance of collagen at the start of the introduction section.

2.      How do healthcare providers diagnose osteogenesis imperfecta (OI)?

3.      How do bisphosphonates work or their mechanism of action? Describe the side effects.

4.      Please include the chemical structure of zoledronic acid (ZA) that would be helpful to understand the mechanism.

5.      The authors did not evaluate the serum levels of calcium, parathyroid hormone (PTH), and vitamin D metabolites in context with zoledronic acid (ZA) treatment of osteogenesis imperfecta (OI).

6.      Include the principle and applications of resonance Raman spectroscopy in the methodology section?

7.      The concept of pentosidine as a biomarker of bone quality and bone fragility has been used in the determination of osteogenesis imperfecta (OI). Pentosidine formation in the body requires glycation and oxidation of proteins. In bone, the protein that is most affected is type 1 collagen by glycation. It is advised to briefly describe glycation, AGEs formation, and the significance of protein glycation in the discussion section to correlate the importance of pentosidine in OI pathogenesis.

Cite the following publication.

a. A structural study on the protection of glycation of superoxide dismutase by thymoquinone. https://doi.org/10.1016/j.ijbiomac.2014.06.003.

b. A review on mechanism of inhibition of advanced glycation end products formation by plant derived polyphenolic compounds. https://doi.org/10.1007/s11033-020-06084-0.

8.      The authors considered only the positive effects of ZA on bone microarchitecture and mechanical properties. The authors did not discuss the adverse effects of zoledronic acid.

9.      Conclusion is very short. The conclusion should be rewritten in a separate section with a brief discussion of results and future perspectives.

10.  Significance of this study should be provided at the end of the introduction section.

11.  Considering the observed effects, I suggest a paragraph at the end of the discussion that the observed effects of the ZA would be an excellent way to treat OI.

12.  Some introductory lines should be written at the beginning of the Results section to better link the different typologies of Results.

13.  The authors did not discuss the name of the mutated amino acid residue in collagen in OI and the mechanism of mutation.

14.  The table ligands should be more informative.

15.  The manuscript should be strictly checked for grammar. Some sentences are not grammatically sound.

1. The manuscript should be strictly checked for grammar. Some sentences are not grammatically sound.

Round 2

Reviewer 1 Report

Thanks to the author for the sufficient revision, I am satisfied with the result of the revision

Good

Reviewer 2 Report

I have no further comments on this manuscript.

It's now fine for publication.